# Plant-Derived Extracellular Vesicles and Their Exciting Potential as the Future of Next-Generation Drug Delivery

**DOI:** 10.3390/biom13050839

**Published:** 2023-05-15

**Authors:** Faisal A. Alzahrani, Mohammad Imran Khan, Nader Kameli, Elham Alsahafi, Yasir Mohamed Riza

**Affiliations:** 1Department of Biochemistry, Faculty of science, Embryonic Stem Cell Unit, King Fahad Center for Medical Research, King Abdulaziz University, Jeddah 21589, Saudi Arabia; faahalzahrani@kau.edu.sa; 2Centre of Artificial Intelligence for Precision Medicines, King Abdulaziz University, Jeddah 21589, Saudi Arabia; mikhan@kau.edu.sa; 3Department of Medical Laboratory Technology, Faculty of Applied Medical Sciences, Jazan University, Jazan 82621, Saudi Arabia; nakameli@jazanu.edu.sa; 4Medical Research Center, Jazan University, Jazan 45142, Saudi Arabia; 5Department of Basic and Clinical Sciences, Faculty of Dentistry, Umm AlQura University, P.O. Box 715, Mecca 21955, Saudi Arabia; ensahafi@uqu.edu.sa

**Keywords:** plant-derived extracellular vesicles, next-generation drug delivery, toxicology, commercial viability, PDEV task force

## Abstract

Plant cells release tiny membranous vesicles called extracellular vesicles (EVs), which are rich in lipids, proteins, nucleic acids, and pharmacologically active compounds. These plant-derived EVs (PDEVs) are safe and easily extractable and have been shown to have therapeutic effects against inflammation, cancer, bacteria, and aging. They have shown promise in preventing or treating colitis, cancer, alcoholic liver disease, and even COVID-19. PDEVs can also be used as natural carriers for small-molecule drugs and nucleic acids through various administration routes such as oral, transdermal, or injection. The unique advantages of PDEVs make them highly competitive in clinical applications and preventive healthcare products in the future. This review covers the latest methods for isolating and characterizing PDEVs, their applications in disease prevention and treatment, and their potential as a new drug carrier, with special attention to their commercial viability and toxicological profile, as the future of nanomedicine therapeutics. This review champions the formation of a new task force specializing in PDEVs to address a global need for rigor and standardization in PDEV research.

## 1. Introduction

Extracellular vesicles (EVs) are unique lipid-bound membrane vesicles secreted in both plant and animal cells that enclose unique and powerful biochemically active compounds such as miRNA, lipids, proteins, and small biomolecules generated through the multivesicular body pathway or from the plasma membrane to the extracellular space [1]. Their size from animal origin is typically divided into four main categories of extracellular vesicles that result from different processes of formation, and they can be roughly differentiated based on their size. These categories are exosomes (50–150 nm), microvesicles (100–1000 nm), large oncosomes (1000–10,000 nm), and apoptotic bodies (100–5000 nm) [2]. However, recently, researchers have been able to isolate and characterize EVs of plant origin that are typically in the range of 30–500 nm [3,4], with some EVs isolated from carrots being as large as 1500 nm [5].

The differences between plant- and animal-derived EVs do not end there, as there are clear and startling differences. PDEVs were found to have natural anticancer, anti-tumor, antioxidant, anti-inflammatory, and wound-healing effects, among many others, from a seemingly limitless plethora of fruits and vegetables, each with their own unique and powerful therapeutic effects. It is noteworthy that ginger- and grapefruit-derived EVs have been studied extensively [6]. These innate natural therapeutic properties can be directly translated to human health, as not only are these effects beneficial, but there have been no reported adverse effects [7].

Interestingly, they were also found to play an important role in inter-kingdom communication that contributes to the symbiotic relationship between plants and micro-organisms and a host of other mechanisms [8,9]. In addition, compared to chemically synthesized article nanocarriers, plant EVs are not only safe, non-immunogenic, and non-toxic [10] but, fascinatingly, they can pass through the blood-brain barrier, while being able to deliver drugs to the pregnant mother without affecting the fetus [10], making it a game-changing avenue for the future of drug design. 

Limitations of chemically synthesized nanocarriers and animal-derived EVs in drug delivery include, but are not limited to, low drug loading efficiency, low yield, short bioavailability due to short half-life in circulation, limited target tissue uptake efficiency, biocompatibility or immunogenicity issues, safety and ethical concerns (for EVs derived from animal sources, including human sources), and being unsustainable and unscalable for large-scale production [11,12,13].

Plant EVs by comparison can be scaled up economically, while being stable even within the gastrointestinal tract, carrying a variety of cargo loads, such as siRNA, miRNA, and even CRISPR/Cas9, with efficient absorption, and therefore, their commercial viability is much more promising than chemically synthesized or animal origin nanocarriers [14]. 

At present, several clinical trials are currently being undertaken to develop drug delivery systems for clinical diagnosis and treatment of diseases using EVs derived from plant and animal sources, including, but not limited to, the use of EVs as biomarkers, drug delivery carriers, cancer vaccines, and cell-free therapeutic agents, including use in regenerative medicine [12,15]. The most significant limitation of EVs as a next-generation drug delivery system is the need to generate a large-scale production of biocompatible EVs for commercial viability. Animal EVs are difficult to scale up due to the limitations related specifically to their isolation from live sources and usually from difficult to procure areas such as mesenchymal stem cells (MSCs), whereas PDEVs by comparison are much easier to scale up due to their relative ease of isolation and procurement of several plant and fruit sources in the form of fresh fruit or juice extracts and inherent advantageous features.

This review endeavors to summarize the latest advances in PDEVs, specifically its role as a nanocarrier of various drugs for the treatment of diseases (see Figure 1), as growing evidence suggests that PDEVs will be the future in a variety of fields, especially in the pharmaceutical and biotechnological industries.

## 2. Composition of PDEVs

### 2.1. Structure 

Structurally, animal- and plant-derived EVs are similar in nature. They are both membranous structures constructed of lipid bilayers, loaded with a variety of biochemically active compounds and substances such as nucleic acids, lipids, and proteins. 

In terms of particle size, PDEVs have been found to be typically similar compared to animal EVs and range in size between 30 and 1500 nm, displaying a high level of stability, with a zeta potential that is generally negative and above the range of −20 mV [4,16,17,18,19].

### 2.2. Composition

PDEVs contain unique cargo loads depending on the type and source of plant or fruit, which in turn determines their biochemical functionality, natural innate therapeutic properties, bioavailability, and related processes. 

The most important and well-studied of these exosomal cargo loads is miRNAs, which are short non-coding RNAs of an average size of about 22 nucleotides (nt), capable of regulating gene expression by specifically binding and inhibiting translation function or cleaving of target mRNAs [20]. Inter-kingdom communication and the functionality of PDEVs have been attributed to miRNAs. PDEVs have been shown to contain and deliver functionally active miRNAs from dietary sources and are capable of surviving the harsh environment of the digestive tract [21,22]. It was shown that not only are they able to survive and be absorbed but they are also capable of traveling to neighboring or distant regions of the body to regulate gene expression of humans, which may indicate a pronounced effect on the role of dietary EVs of plant origin on the health and physiochemical function and regulation of human health and nutrition [23]. Several studies have shown that EVs derived from a plethora of plants of various species have been implicated to have miRNAs that target genes associated with several cancer- and inflammation-related pathways. Of note, ginger-derived miRNAs were found to inhibit lung inflammation caused by EVs released from SARS-CoV-2 [24].

As for studies on the protein content of PDEVs, it is severely limited due to the lack of proper databases of proteomic analysis, contaminations from cytosolic or enzyme proteins, and most importantly, their typically low protein concentration; however, it was seen that this can change significantly in response to biotic and abiotic stresses such as external infections by pathogens [25,26,27].

In contrast, lipids, specifically phospholipids, derived from PDEVs are well studied and known to play a vital role in plant exosomal stability and the uptake by specific targeting of cells, such as the gut and the liver, and their lipid composition affects which recipient cells are targeted and thereby affected by its cargo [28,29,30]. This is in contrast to lipids of animal origin, as they are predominantly sphingomyelin and cholesterol [1]. It has also been shown that nanocarriers derived from plant-origin exosomal lipids have been used with resounding success for the delivery of siRNA and anticancer agents [10].

It has been shown that PDEVs even carry secondary metabolites, such as flavonoids, chlorophylls, and curcuminoids, which were found to differ based on the origin of the plant/fruit species [10,27,28]. These secondary metabolites have been linked to their potential antimicrobial therapeutic activity and shown to increase in concentration seemingly in response to infections from pathogenic organisms [27,28].

## 3. Isolation and Characterization 

For commercial viability of large-scale production of nanocarriers made using PDEVs, we need a method of isolation and purification that is time- and cost-effective, with a balanced level of relative purity and yield. There are currently several methods available, but there is still no gold standard for the ideal isolation method for large-scale commercial production. Each method has its own advantages and disadvantages, so their use and relevance are dependent on their downstream applications. Therefore, our review gives the perspective of a time-efficient and cost-effective method for large-scale production.

Ultracentrifugation, used typically along with a sucrose gradient medium, is the most commonly used method that generates EVs with a high level of purity with a relatively simple operation; however, it is substantially tedious and takes a significantly long time, besides being one of the costlier methods of isolation [3,30]. Currently, commercial viability does not seem promising, as it is very difficult to scale up, apart from being unsustainable.

Electrophoresis *when* combined with dialysis, on the other hand, is its opposite, where it offers the advantages of being convenient, with a high yield and a relatively low cost, but suffers from having a low temperature condition with relatively low levels of purity [3,31]. Commercial viability may be promising if combined with a downstream method of further purification and appropriate long-term storage conditions.

Precipitation with polyethylene glycol (PEG) is a crude and non-specific technique that separates proteins by virtue of their solubility. PEG acts as an inert solvent sponge, reducing solvent availability. This method offers a simple enough operation while remaining economical and providing a rapid rate of isolation; however, it offers relatively lower levels of low yield and purity [3,32,33,34]. Commercial viability may be promising if yield levels can be improved or at least concentrated downstream, possibly in combination with a downstream method of further purification.

Arguably, size exclusion chromatography (SEC) is one of the best methods of isolation as it *provides* a very high level of yield with relatively high levels of purity, while also providing EVs with good levels of functionally active bioactivity. However, it suffers from an extremely taxing workload that takes a significantly long time, making commercial viability unlikely unless it can be scaled up effectively [3,35].

Commercially available EV isolation test kits provide a convenient and easy method for EV isolation, but often yield low purity levels and are typically used on a small scale [3,36]. However, some manufacturers claim that their higher-end kits can provide up to 420 times more EV isolation compared to ultracentrifugation, resulting in relatively higher yields, higher purity levels, and better biomarker detection levels. These kits can isolate EVs in less than 20 min and are relatively cost-effective [37]. This might make it the most commercially viable method if it could be scaled up appropriately, as it provides a good balance between all the requirements (see Figure 2). 

To summarize, the outcomes of functional studies performed to evaluate EV purity, possibility of contamination, and relative yields of cargo contents paint a picture of balancing varying levels of disadvantages depending on your needs and downstream applications.

Differential centrifugation can be followed by density gradient ultracentrifugation to separate low-density EVs from high-density protein aggregates that often contaminate EV isolates. Size exclusion chromatography (SEC) is now widely used for low-volume samples and allows separation of EV from soluble proteins, but may co-isolate other particles in the EV size range, such as (lipo)protein complexes. Immuno-affinity capture can yield pure EV subpopulations, but the choice of affinity reagent and ligand density on different EV types can influence the results. Commercial kits that use volume-excluding polymers, such as polyethylene glycol (PEG), are available for rapid EV isolation, but these polymers also co-precipitate protein (complexes) that contaminate EV isolates. Different methods will enrich different subpopulations of vesicles and co-isolate contaminants to varying degrees [38,39,40]. 

Combinations of techniques are often used, and comparative studies on different techniques have been published. Limited data are available on the impact of different EV isolation methods on EV-RNA yield and purity. Some isolation techniques give higher yields of RNA but come at the cost of lower purity, which can affect the conclusions drawn from RNA analysis. The optimal isolation method and acceptable impurities depend on the research question and downstream analysis, the type and volume of the starting material, the number of samples, and the laboratory infrastructure. In the discovery phase of EV-RNA biomarkers, contamination of EV isolates may lead to erroneous conclusions, and pure EV populations are required for studies on the role of EV-RNA in physiological processes. However, in detecting established EV-RNA-based biomarkers in low-volume patient samples, increased protein/lipid/RNA yields at the cost of lower EV purity may be acceptable [38,39,40].

It is important to note that since there is a plethora of potential PDEVs, most of which have not been properly studied and understood, the source of the EVs and their relative origins (depending on the region or country *they have been obtained from*) may affect the ideal type of EV isolation method. Most of the functional studies currently in the field relate to animal-derived EVs. 

EVs are obtained from a variety of sources, including body fluids with different compositions, cell culture media, and tissues from different species. It is difficult to provide general recommendations for EV isolation and characterization due to the variability in sample sources. Although previous recommendations for EV isolation from some body fluids still hold, recent research has provided a better understanding of contaminants in EV isolates from different body fluids and the pre-analytic variables that affect EV purity. Some body fluids may contain specific contaminants, such as bacteria-derived material in nasal fluid, saliva, milk, and urine, or biofluid-specific contaminants in synovial fluid samples. Standardized pre-analytical conditions for EV isolation should be determined for each biological fluid separately. For example, *blood plasma* has been used as a source of EVs, but its complexity presents challenges for EV isolation in RNA analysis studies. Recent developments in EV research in plasma illustrate the difficulties encountered when performing EV-RNA analysis in body fluids [38,39,40].

Overall, the ideal method of EV isolation does not currently exist, and perhaps a combination of several methods while balancing the disadvantages of one with the advantages of the other might be the only realistic outcome while balancing cost and time requirements.

## 4. Physical Characterization

In order to characterize whether EVs have been successfully isolated requires appropriate analysis of their physio-structural properties, such as their size, zeta potential, morphological characteristics, and so on. This can be achieved using nanoparticle tracking analysis (NTA) [41] or other methods, including transmission electron microscopy (TEM) [42] and scanning electron microscopy (SEM) [43], as well as dynamic light scattering (DLS) [42], atomic force microscopy (AFM) [42], and tunable resistive pulse sensing (TRPS) [40,44]. 

Each of these methods are relatively comparable to each other, as they all allow for the determination of stable and authentic nano-size EVs of plant origin. Currently, for EV 1characterization, NTA is considered the gold standard [45].

## 5. Minimal Information for Studies of Extracellular Vesicles 2018 (MISEV2018)

The MISEV2018 Checklist is a set of guidelines developed by the International Society for Extracellular Vesicles (ISEV) to standardize the reporting of studies involving extracellular vesicles (EVs). *Our review endeavors to briefly summarize the guidelines in order to raise awareness among researchers about the importance of adhering to the guidelines recommended by experts.*


EVs are small membrane-bound structures that are secreted by cells and that play important roles in cell-to-cell communication and signaling. The guidelines provide a checklist of information that should be included in studies of EVs to ensure that they are accurately characterized and properly studied. The guidelines are divided into six sections: nomenclature, collection and pre-processing, EV separation and concentration, EV characterization, functional studies, and reporting.

In the nomenclature section, the guidelines require the use of the generic term “extracellular vesicle” (EV) to describe the vesicular nature of the structure, with further specification of the subcellular origin of the vesicle, if known.

The collection and pre-processing section of the guidelines provides detailed information on the methods used to collect and process EVs, including the nature and size of culture vessels, volume of the medium during conditioning, and frequency of the conditioned medium harvest. For biofluids and tissues, the guidelines require the reporting of donor status and all known collection conditions, including additives, at the time of collection.

The EV separation and concentration section of the guidelines requires the experimental details of the method used, including centrifugation, density gradient, chromatography, precipitation, filtration, antibody-based, and other methods. The guidelines also require the specification of the category of the chosen EV separation/concentration method, based on the level of recovery and specificity of the isolated EVs.

The EV characterization section of the guidelines requires the reporting of the global quantification of EVs by at least two methods, such as protein amount, particle number, and lipid amount. The guidelines also require the global characterization of EVs based on their size, shape, surface markers, and cargo molecules. In addition, the MISEV guidelines recommend assessing the presence or absence of expected contaminants, including at least one of each of the three categories of contaminants. It is also recommended to investigate the presence of proteins associated with compartments other than the plasma membrane or endosomes, as well as soluble secreted proteins and their likely transmembrane ligands.

In terms of EV characterization, the guidelines suggest performing a single EV characterization, including the use of various imaging techniques, such as electron microscopy, scanning probe microscopy, and super-resolution fluorescence microscopy, to obtain wide-field and close-up images of single EVs. Additionally, non-image-based methods for analyzing large numbers of single EVs, such as NTA, high-resolution flow cytometry, multi-angle light scattering, and Raman spectroscopy, should also be considered.

For functional studies, the guidelines recommend conducting a dose-response assessment using a negative control, such as a nonconditioned medium, biofluid/tissue from control donors, or something similar. It is also recommended to quantitatively compare the functional activity of total fluid, EV-depleted fluid, and EVs after high recovery/low specificity separation, as well as to compare the functional activity of EVs with other EPs/fractions after low recovery/high specificity separation. If subtype-specific function is claimed, a quantitative comparison of the activity of EV subtypes should be conducted. Finally, the guidelines suggest investigating the extent of functional activity in the absence of contact between the EV donor and the recipient.

In terms of reporting, the MISEV guidelines strongly encourage submitting methodological details to EV-TRACK with an EV-TRACK number provided. Data should also be submitted to relevant public or curated databases, or open-access repositories, including EV-specific databases such as EVpedia, Vesiclepedia, and exRNA atlas. The guidelines recommend tempering EV-specific claims when MISEV requirements cannot be entirely satisfied.

Overall, the MISEV2018 Checklist provides a standardized approach for reporting studies of EVs, which can help ensure that studies are accurately characterized and can aid in the comparison of results across studies [46].

PDEVs are a relatively new area of study, and there is still much to be learned about their characteristics and functions. Since the MISEV guidelines do not specifically address PDEVs, researchers should strive to adhere to the expert-recommended guidelines and report as much detail as possible to standardize the collection and pre-processing of PDEVs.

However, based on the authors’ expert discourse on the subject, here are some suggestions for important caveats to be considered for standardization in the collection and pre-processing of PDEVs that might be useful as an input to the development of the next MISEV guidelines:Characterization of plant material: The guidelines should require reporting of the botanical identification, origin, and growth stages of the plant material. This will help ensure that the same plant material is being used consistently across different studies.Collection of plant material: The guidelines should require reporting of the method of collection, including whether the plant material was collected from a greenhouse or field, and whether the plant material was harvested in the morning or evening.Processing of plant material: The guidelines should require reporting of the method of processing, including the type of buffer used for extraction, the method of homogenization, and the duration of extraction.Purification of PDEVs: The guidelines should require reporting of the method of purification, including the type of ultracentrifugation or other isolation method used, and the duration of centrifugation.Characterization of PDEVs: The guidelines should require reporting of the method of characterization, including the use of imaging techniques such as electron microscopy, as well as the quantification of PDEVs based on protein amount, particle number, and lipid amount.Functional studies of PDEVs: The guidelines should recommend conducting functional studies to investigate the biological effects of PDEVs, including their ability to transfer cargo molecules to recipient cells, and their potential use as therapeutic agents.Reporting of data: The guidelines should require the submission of data to relevant public databases or open-access repositories, including EV-specific databases such as EVpedia, Vesiclepedia, and exRNA atlas.

Additionally, it would be beneficial for researchers to compare their PDEVs to known EVs derived from other sources, such as mammalian cells, to ensure that their PDEVs meet the criteria for EVs, such as the size, morphology, and expression of EV markers.

Moreover, it is important to consider the potential effects of plant-specific compounds, such as polyphenols and flavonoids, on the isolation and characterization of PDEVs. These compounds may affect the stability of PDEVs and the accuracy of characterization methods. Therefore, it is recommended that researchers investigate the potential effects of these compounds and report any relevant findings.

Furthermore, an important caveat to note is that biotic and abiotic stress on plants can affect the composition and characteristics of PDEVs. Therefore, it is important to consider the stress conditions when standardizing the collection and pre-processing of PDEVs. When collecting plant material, researchers should report whether the plants were subjected to any stress conditions, such as drought, heat, cold, or pathogen infection. This information can help identify any changes in PDEVs that may be related to stress. In terms of pre-processing, it may be necessary to modify the extraction and purification methods to accommodate changes in PDEVs under stress conditions. For example, if stress conditions lead to an increase in the number of certain types of PDEVs, it may be necessary to adjust the extraction protocol to ensure that these PDEVs are properly isolated. Overall, the standardization of the collection and pre-processing of PDEVs should take into account any biotic or abiotic stress that the plants may have been subjected to, and researchers should report this information in their methods section.

Finally, as PDEVs are a relatively new area of study, there is a need for continued research and refinement of methods for their collection, processing, and characterization. Researchers should stay informed of the latest developments in the field and adapt their methods accordingly to ensure the most accurate and reliable results. There is a global need for a task force focused on PDEVs that standardizes and disseminates the latest information and best practices for researchers in the field. We recommend the formation of a new task force by the International Society of Extracellular Vesicles and hope that our review would champion such an effort.

## 6. Bioactivity 

Plant-based treatments for illnesses have been utilized for a significant amount of time, utilizing either their natural forms or extracts of active compounds, such as polysaccharides, phenols, and terpenoids, to prevent illness and repair harm caused by disease [47,48]. Nevertheless, only recently have researchers started to examine PDEVs, a new plant component. 

Through initial investigation, PDEVs from various edible plants have demonstrated strong biological properties, such as anti-inflammatory, anticancer, antibacterial, *antifungal*, and antioxidative effects [3,6,14,23,27,30]. They produce these effects through various mechanisms, such as gene regulation, influence on gut microbiota, macrophage activity, gene silencing, and the presence of specific active molecules [1,29,36].

These innate therapeutic properties are a promising avenue, whether used alone or in combination with other therapeutic agents and nanomedicines.

### 6.1. Anticancer Effect

The treatment of cancer has a long history, and numerous methods and drugs have been developed to combat malignant tumors. However, due to the resistance of malignant tumors and the adverse effects of drugs, an ideal solution has yet to be found. While many drugs are available for treating cancer, their non-specific impact on the immune system can cause short-term and long-term detrimental side effects, such as myelosuppression, cardiovascular adverse effects, *including* cardiotoxicity and hypertension, and other potentially fatal consequences [49].

On the other hand, PDEVs have minimal side effects and have demonstrated varying degrees of therapeutic effects against various types of cancer. According to research, EVs derived from plants such as lemon, ginger, grape, grapefruit, and Chinese bamboo shoots all possess anticancer properties [50,51]. Lemon-derived EVs have an anti-proliferative effect in both in vivo and in vitro environments. They can increase the expression of GADD45a through the reactive oxygen species (ROS) generated by the tumor tissue and cause S-phase arrest and apoptosis in gastric cancer cells [31]. 

EVs from citrus, lemon, and grape inhibit the growth of A375 (melanoma), A549 (lung adenocarcinoma), and MCF-7 (breast cancer) cell lines to varying degrees [52]. Tea flower-derived EVs, after intravenous injection or oral administration, accumulate in breast tumors and lung metastatic sites, inhibit the growth and spread of breast cancer, and regulate the gut microbiome [53]. Lemon-derived EVs can also specifically reach the tumor site and activate the TRAIL-mediated apoptotic cell process, inhibiting chronic myeloid leukemia in vivo [54]. *Asparagus cochinchinensis*-derived EVs exhibit specific anti-proliferative activity against hepatocellular carcinoma cells and induce apoptosis through a related pathway [50].

The safety, effectiveness, targeting, and anti-proliferative capabilities of PDEVs, *which may be used by itself or potentially* loaded with FDA-approved anticancer drugs *that are able to fit as a cargo within them*, may offer a new avenue for the synergistical prevention and treatment of cancer. *However*, further research is needed to provide a conclusive outcome. 

### 6.2. Antioxidative Action

Oxidation leads to aging, inflammation, and other negative outcomes. PDEVs have a lipid bilayer structure that helps preserve unstable antioxidants within vesicles. Antioxidants found in fruits and vegetables with high antioxidant activity can be encapsulated in their corresponding EVs and transported to plasmids in a laboratory setting. 

Research has shown that lemon-derived EVs contain high levels of citric acid and vitamin C, which effectively protect mesenchymal stem cells from oxidative stress [55]. Strawberry-derived EVs can be taken up by MSC without altering their activity and are able to prevent oxidative stress in a dose-dependent manner, likely due to the high levels of vitamin C in the vesicles [17]. 

The antioxidant properties of PDEVs hold great promise in the fields of cosmetics and medicine. It is important to consider these vesicles as novel ingredients in our food to evaluate their health benefits and potential impact on food technology.

### 6.3. Anti-Inflammatory Action

Inflammation is caused by an imbalance in the immune system and can lead to acute or chronic diseases [56]. PDEVs, such as those from ginger, grapefruit, and grapes, have demonstrated anti-inflammatory properties in several studies [4,6,10]. Both lipids and RNA in these vesicles contribute to their anti-inflammatory effects through different mechanisms [10]. 

Lipids can regulate gene expression in the body’s inflammatory areas, improve inflammation through their actions on macrophages, and contain miRNAs that can change the composition of the gut’s microbiome [10,20,21,22,24]. 

Ginger-derived EVs play a role in anti-inflammation by activating Nrf nuclear translocation and inducing anti-inflammatory cytokines [57], while grapefruit-derived EVs regulate anti-inflammation through Wnt signaling [58,59]. Nrf2 nuclear translocation and Wnt/TCF4 activation are important factors in the anti-inflammatory response [60].

### 6.4. Antimicrobial Effect

Studies on PDEVs and their impact on cell functions have advanced quickly, but their effect on bacterial growth still requires further investigation and a comprehensive summary. Some PDEVs can support the growth of certain probiotics, while others can hinder the growth of harmful bacteria. 

Recent research has revealed that some PDEVs can be absorbed by bacteria when they are co-incubated under appropriate conditions [60]. The internal miRNAs in EVs may play a role in regulating bacterial growth, such as the promotion of probiotics in coconut water [19]. EVs from *Arabidopsis thaliana* can transfer small RNA to the site of fungal infection, where they can be absorbed by the pathogen *Botrytis cinerea* and suppress important fungal genes [61]. 

Ginger-derived EVs are specifically taken up by *Porphyromonas gingivalis* through interactions with heme-binding proteins on the bacteria’s surface. The pathogenic capability of *Pseudomonas gingivalis* is significantly weakened after interacting with ginger-derived EVs, which contain *phosphatidic acid* and miRNAs [60]. Plant-root-released EVs with antifungal properties have also been identified [61].

Given the demonstrated ability of PDEVs to affect bacterial growth, there is potential for their use in treating bacterial diseases.

## 7. The Use of PDEVs in Treating Human Illnesses

People are very interested in the potential of PDEVs to play a role in interspecies communication and to have direct benefits for human diseases [62,63,64]. These EVs consist of proteins, RNA, lipids, and other active ingredients that can be absorbed into the intestine through oral consumption and then transported to different parts of the body [65]. Additionally, they contain plant-similar components that have potential biological functions similar to their plant counterparts. PDEVs are able to withstand the harsh acidity in the stomach and the highly active proteolytic enzymes in the intestine [66]. 

Studies have shown that edible PDEVs can be absorbed into the intestine through oral consumption, regulating intestinal flora or inflammatory factors, affecting intestinal stem cells, or regulating downstream gene expression through small RNA (sRNA) [6,57]. Researchers have applied PDEVs to animal models in various ways, such as oral, intravenous, nasal, or transdermal administration. However, oral administration is considered to be a relatively safe route for safety reasons [3,6]. 

The appropriate administration route for PDEVs will vary depending on the needs of the disease. Additionally, EVs from the same plant source can be administered in various ways to treat different diseases. They have been found to be safer for treating animal models than drugs with strong side effects, such as in the treatment of colitis and alcoholic liver.

sRNA, a type of non-coding regulatory RNA that ranges from 20 to 30 nucleotides, plays a crucial role in controlling gene expression [67,68]. sRNA can spread to different parts of animals through EVs, transmembrane proteins, high-density lipoprotein complexes, or gap junctions [69]. 

In plants, sRNA can be transmitted from one cell to another through plasmodesmata or the vascular system [70]. Recent research has shown that sRNAs can also be transferred between different eukaryotic species, connecting the world of animals, plants, and microbes [68]. 

There is evidence of various types of sRNA present in PDEVs, which can be carried and introduced into eukaryotic cells through EVs, leading to a therapeutic effect through RNA interference through various mechanisms of action [57,61,71]. 

### 7.1. Change the Composition of the Gut Microbiota and Affect the Physiology of the Host

Revising gut bacteria is a novel and efficient approach to treating colitis. PDEVs are absorbed by the gut microbiota and contain RNA that changes both the composition of gut bacteria and host physiology. 

Evidence suggests that plant products and their impact on the gut microbiota can be utilized to alleviate diseases that affect certain host processes [3,57]. Ginger-based EVs, for example, contain miRNAs that target various genes in *Lactobacillus rhamnosus* and are taken up preferentially by *Lactobacillaceae* through a lipid-dependent mechanism. After being taken up by *Lactobacillus rhamnosus*, the miRNAs in ginger-based EVs target monooxygenase ycnE and increase the production of indole-3-carboxaldehyde. 

The RNA in ginger-based EVs and indole-3-carboxaldehyde, which is a ligand for the arylhydrocarbon receptor, are enough to stimulate the production of IL-22, which is associated with improved barrier function. These properties of ginger-based EV-RNA can improve mouse colitis through IL-22-dependent mechanisms [29].

### 7.2. Modifying the Activity of Intestinal Macrophages

Intestinal macrophages are critical for maintaining the stability of the host by preserving mucosal tolerance and preventing inflammation. However, during the pathophysiology of inflammatory bowel disease (IBD), these macrophages lose their tolerance, leading to uncontrolled intestinal inflammation. Regulating the function of these macrophages is therefore crucial for treating IBD patients [10].

Studies have demonstrated that grapefruit-derived EVs can selectively target and improve the function of intestinal macrophages in mice with colitis. These EVs are taken up by the macrophages and increase the expression of the anti-inflammatory protein HO-1, while decreasing the production of pro-inflammatory cytokines, such as IL-1β and TNF-α. Additionally, grapefruit-derived EVs are considered safer than traditional treatments such as immunosuppressants and steroids due to their edible plant origin, biocompatibility, biodegradability, and ability to withstand a wide range of pH levels [10].

Furthermore, grapefruit-derived EVs have also shown promising results as drug delivery systems. Incorporating anti-inflammatory drugs into EVs and administering them to mice with colitis has been shown to significantly reduce drug toxicity and improve therapeutic efficacy compared to administering the drugs directly. These findings suggest that grapefruit-derived EVs can be used as an oral drug delivery system to reduce inflammation in human diseases [10].

### 7.3. Combined Therapeutic Potential in the Treatment of Colitis 

The condition of colitis is characterized by changes in the colon that can be caused by various factors, such as genetics, the environment, immune system factors, and infections. The typical symptoms of colitis are abdominal pain, diarrhea, rectal bleeding, and constipation [3]. The traditional treatments for colitis involve the use of high doses of drugs, such as antibiotics, anti-inflammatory medications, biological agents, and immunomodulators, to reduce inflammation [3]. However, these drugs can be toxic if used for a long time and may not be effective in the long run [72]. 

Efforts to find a safer, more effective, and more stable treatment system have been ongoing, and PDEVs have emerged as a potential solution. PDEVs are safe and stable, and they have shown promising results in the treatment of colitis [73]. Different types of EVs derived from different plants contain different lipids and RNA molecules, which give them unique properties and mechanisms of action [64].

### 7.4. Regenerative Properties of PDEVs

The treatment of collagen synthesis can be improved by using PDEVs from *citrus limon*. Citrate, a micronutrient found in citrus juice, is crucial in maintaining the structure of the bone matrix and regulating bone health. The effect of these PDEVs may be due to the presence of citrate and other beneficial molecules [55].

EVs obtained from aloe vera peels were effectively taken up by human keratinocytes (HaCaT) cells through clathrin- and caveolae-mediated endocytosis. The aloe vera vesicles (A-EVs) heightened the antioxidant defense by regulating Nrf2 and reducing oxidative damage in a dose-dependent manner. In vitro wound healing tests have shown that A-EVs can enhance the migration of keratinocytes and fibroblasts to the wound site, indicating their potential for skin regeneration therapy [74]. Research has also revealed that EVs derived from wheat grass juice have a significant impact on the proliferation and migration of endothelial (HUVEC), epithelial (HaCaT), and dermal fibroblast (HDF) cells [75]. 

This highlights the potential for further exploration of EVs from specific plants to develop products with wound-healing and cosmetic applications.

### 7.5. The Therapeutic Impact of PDEVs on COVID-19

The COVID-19 pandemic has caused widespread death and major disruptions to the global economy, yet no effective treatment has been found. PDEVs from dietary sources contain a high amount of small RNA, which has the potential to regulate communication between plants and other species [61]. 

The antiviral potential of ginger-derived EVs was confirmed through sequencing of high-abundance RNA, but further research is needed to study their stability and antiviral activity [76]. In a study on COVID-19 pathogenesis, EVs released by severe acute respiratory syndrome coronavirus type 2 (SARS-CoV-2) cells were found to play a role in the development of pulmonary inflammation, while ginger-derived EVs were shown to inhibit Nsp12 gene expression and reduce the effects of SARS-CoV-2 in the lungs [24]. 

These findings suggest that the RNA contained within PDEVs has potential for the treatment of COVID-19.

### 7.6. Maintenance of the Balance of the Immune System within the Intestines

The regulation of the immune system balance in the gut is maintained by broccoli-derived EVs, primarily affecting dendritic cells (DCs). *The adenosine monophosphate-activated protein kinase (AMPK)* in DCs is a critical enzyme and pathway involved in regulating the immune system balance [77]. AMPK is present in various immune cells, including dendritic cells, macrophages, lymphocytes, neutrophils, and others, and controls their functions, including cytokine production, chemotaxis, apoptosis, cytotoxicity, and proliferation. 

Broccoli-derived EVs can target DCs and activate the AMPK in these cells, which results in the inhibition of DC activation and monocyte recruitment. At the same time, sulforaphane in broccoli-derived EVs stimulates the production of tolerant DCs. The combination of these pathways produces regulatory DCs, which prevent colitis in mice [64]. This study provides evidence that edible PDEVs containing food-specific antigens from plants can induce oral tolerance. Broccoli-derived EVs cause immune DCs or tolerant DCs to form through the epithelial barrier, preserving the balance of the immune system in the gut [3]. 

These results suggest that edible PDEVs, which are naturally targeted in the colon and have anti-inflammatory properties, may offer a new and natural, non-toxic drug delivery system that can be easily expanded to treat gastrointestinal diseases, such as inflammatory bowel disease (IBD) [66].

### 7.7. Ability to Enhance the Growth of Intestinal Stem Cells

Grape-derived EVs can protect against mouse colitis caused by dextran sulfate sodium by increasing the proliferation of intestinal adult mouse *colon* stem cells (Lgr5hi) and by facilitating the remodeling of intestinal tissue in response to harmful stimuli. After oral administration of the grape-derived EVs into the gut, fluorescence showed that they accumulated in the intestine within the first 6 h and then gradually declined, but remained in the gut for up to 48 h. 

The EVs targeted Lgr5hi stem cells in the mouse intestine and encouraged their growth, speeding up the regeneration of the mucosal epithelium. Researchers discovered that after the stem cells ingested the grape-derived EVs, the number of BMI1 genes increased, and the expression of pluripotent stem cell markers, such as SOX2, Naong, OCT4, and KLF4, was significantly elevated. 

This, in turn, activated the downstream Wnt signaling pathway, resulting in the proliferation of Lgr5+ stem cells, faster mucosal regeneration, remodeling of the intestine, protection of the mouse gut, and inhibition of colitis caused by DSS. Furthermore, lipids assembled from grape-derived EVs have been found to play a role in targeting and promoting the growth of colon stem cells [4,63].

### 7.8. The Use of PDEVs for the Treatment of Alcoholic Liver Disease

PDEVs from various sources can target specific cells, resulting in therapeutic effects. Once inside cells, *PD*EVs can regulate crucial hubs such as NRF2 and NLRP3, which play important roles in regulating cellular responses to stress and inflammation and can affect downstream gene expression [78,79]. 

For instance, ginger-derived EVs reach the liver via blood vessels and could have a positive impact on liver disease. The extract from ginger-derived EVs can protect mice from alcohol-induced liver damage by activating NRF2 in the liver and promoting the expression of liver detoxification and antioxidant genes, while decreasing reactive oxygen species production. *Lentinus edodes*-derived EVs can reduce liver injury in mice caused by D-galactosamine/lipopolysaccharide by inhibiting the activation of NLRP3 [80]. 

As a result, oral intake of these PDEVs could potentially be used to treat alcoholic liver disease.

## 8. PDEVs as Next-Generation Drug Carriers for the Treatment of Diseases

PDEVs, which are nanoparticles encased in lipid bilayers and contain various substances, can be used as drug delivery platforms. They can be loaded with small-molecule drugs or nucleic acid drugs without degradation, making them an effective method for preventing drug damage [15]. PDEVs have high bioavailability and can be absorbed in the intestine and penetrate deep into the skin [81]. They can pass through the blood-brain barrier and enter the brain when administered nasally [81], while being unable to pass from the mother to the fetus through the placenta [82]. 

PDEVs are also considered safe, as they have low toxicity and fewer side effects compared to synthetic lipid nanoparticles. The current method of drug delivery using PDEVs involves loading drugs onto the EVs or nanoparticles made from the extracted lipid components of the EVs. The majority of drugs used are nucleic acid drugs and small-molecule chemical drugs [83].

### 8.1. Capability of Delivering Nucleic Acids 

PDEVs are capable of delivering nucleic acids, such as siRNA and miRNA. For example, grapefruit-derived EVs loaded with miR17 have been shown to be effective in treating brain tumors in mice by delaying their growth [84]. 

This is because EVs can reach the brain through nasal administration, allowing the miR17 they carry to also enter the brain and exhibit an inhibitory effect. Additionally, EVs combined with folic acid have been found to enhance the targeting of cytomegalovirus to folate receptor-positive brain tumors, providing a non-invasive treatment option for brain-related diseases [84]. 

There are multiple types of PDEVs that can be utilized for a range of delivery pathways. For example, acerola-derived EVs have been found to deliver small RNA to the digestive system, with the target gene-suppressing effect in the small intestine and liver peaking a day after administration, making it a potential option for oral delivery of nucleic acids [85]. Ginger-derived EVs loaded with siRNA-CD98 have also been found to effectively target colon tissue and reduce the expression of CD98 through oral administration [86]. 

Several methods can be used to introduce exogenous nucleic acids into plant-derived extracellular vesicles (PDEVs) [85,87,88,89,90]. Some of the common methods include:Electroporation: This involves applying an electric field to PDEVs and exogenous nucleic acids, causing transient pores to form in the membrane and allowing the nucleic acids to enter the PDEVs.Sonication: This method involves exposing PDEVs and nucleic acids to high-frequency sound waves, which disrupt the vesicle membrane and enable nucleic acids to enter the PDEVs.Incubation: PDEVs can be incubated with nucleic acids in a buffer solution under controlled conditions, such as temperature, pH, and salt concentration. This method allows nucleic acids to be passively taken up by the PDEVs.Extrusion: This involves forcing PDEVs and nucleic acids through a membrane with small pores, which mechanically disrupts the vesicle membrane and enables the nucleic acids to enter the PDEVs.Chemical transfection: This involves treating PDEVs and nucleic acids with chemicals that increase the permeability of the vesicle membrane, allowing the nucleic acids to enter the PDEVs.

It is worth noting that the efficacy and specificity of these methods may vary depending on the type of nucleic acid and PDEVs used, as well as the experimental conditions.

These studies demonstrate that PDEVs can serve as a drug delivery platform for small-molecule drugs, offering further potential applications.

### 8.2. Capability of Delivering Small Molecules and Drugs 

PDEVs can be taken orally and have different cellular targeting due to their different sources [91]. They can also reach the brain via nasal delivery and are intercepted by the placental barrier, making them a highly effective drug delivery option. Grapefruit-derived EVs are capable of transporting drugs, such as chemotherapy agents, nucleic acids, and proteins, to various cells [82]. 

The targeting efficiency of cells that express folate receptors has been improved through the co-delivery of therapeutic agents and folic acid with *grapefruit-derived EVs*. The efficacy of *grapefruit-derived EVs* in inhibiting tumor growth through chemotherapy was demonstrated in two different animal models of tumors. Grapefruit-derived EVs are less toxic than nanoparticles made from synthetic lipids and do not cross the placental barrier when administered intravenously to pregnant mice, making them a promising tool for drug delivery [82].

Grapefruit-derived EVs possess several properties that make them suitable for developing an oral drug delivery system. These properties include biocompatibility, biodegradability, stability in a wide range of pH, and the ability to target specific cells. When combined with the anti-inflammatory drug methotrexate (MTX), the toxicity of the *MTX-grapefruit-derived EVs* was found to be lower compared to free MTX, and the therapeutic effect on mice with DSS-induced colitis was enhanced. This suggests that *grapefruit-derived EVs* could be used as an intestinal immunomodulator and be developed for oral administration of small-molecule drugs to reduce the inflammatory response in human diseases [10]. 

Ginger-derived EVs can also improve targeting by combining with folic acid, and when loaded with the chemotherapy drug doxorubicin, they showed better efficacy in inhibiting tumors and good biocompatibility at a concentration of 200 μmol/L [6]. Ginger-derived EVs also have better pH-dependent drug release properties compared to commercial liposome doxorubicin [73].

Overall, research has shown that using PDEVs for drug delivery offers many benefits, such as safety and efficacy. Because these vesicles come from natural food sources, they have already been demonstrated to be non-toxic in humans. Moreover, compared to other delivery methods, PDEVs appear to have fewer complications and allow for more specific targeting of cells. Several plant-derived vesicles, including ginger and grapefruit, have been shown to have interesting implications for the delivery of microbial agents and the treatment of intestinal dysbiosis. Additionally, plant-derived vesicles have been found to be incapable of passing the placental barrier, making them promising for drug delivery in pregnant mothers. This evidence suggests that plant vesicles have a wide range of possibilities for future use in health and therapeutics, including the treatment of cancer and other serious diseases. However, more research is needed to explore other edible plant sources for EVs [92].

## 9. The Distribution and Uptake of Plant-Based EVs in the Body

In experiments with PDEVs on cells, it has been found that they can be taken in by various animal cells. When PKH-labeled cabbage EVs were incubated with human HaCaT cells for 15 min at 25 °C, the fluorescence showed that the EVs had been absorbed into the cells [35]. Apple-derived EVs were co-incubated with Caco-2 cells and internalized within 6 h [16]. The galactose groups on the surface of tea-derived EVs can be internalized by macrophages via endocytosis, as a result of their binding to galactose receptors [93]. 

Researchers have conducted preliminary studies on the internalization mechanism of PDEVs, which is linked to the proteins on their surface. When the proteins on the surface of garlic-derived EVs were eliminated through digestion with trypsin, the number of garlic-derived EVs taken up by HepG2 cells also decreased. The proteins found to be responsible for this effect include CD68 and II lectin, a protein that binds specifically to mannose [94]. 

To further understand the internalization of *Asparagus cochinchinensis*-derived EVs, HepG2 cells were incubated with endocytosis inhibitors. The uptake of A*sparagus cochinchinensis*-derived EVs was significantly inhibited by cytochalasin D, an inhibitor of actin polymerization that is required for phagocytosis, suggesting that the internalization of EVs likely occurs via the phagocytosis pathway [50,95]. 

Different EVs from different plants have varying affinities for different organs and tissues once they enter mammals. For example, broccoli-derived EVs regulate immune homeostasis in the intestine by targeting DCs [64], while ginger-derived EVs effectively target the colon after oral administration and remain in the stomach, ileum, and colon 12 h after being administered orally [6]. 

The distribution of PDEVs in the body can be related to their particle size, surface charge, and the type and number of membrane proteins, as tea-derived EVs, for example, accumulate in liver tissue due to their galactose-mediated receptor targeting and uptake [93].

## 10. Summary of the Overall Safety, Toxicological Profile, and Biocompatibility of Plant-Sourced EVs

The toxicity of PDEVs (EVs) is low, according to previous studies [6]. For instance, when cabbage-derived EVs were mixed with human and mouse cells for 72 h, there was no substantial decrease in cell survival rate, but rather an increase in cell proliferation was observed [35]. 

Strawberry-derived EVs can be taken up by human bone marrow mesenchymal stem cells without any negative effects and may even provide protection against oxidative stress [17]. Given its low toxicity, PDEVs are considered a promising and safe delivery vehicle for various applications. Of significance is the fact that there have been no reported adverse effects of plant-sourced EVs compared to those that are present in synthetic or animal-derived EVs [7]. 

Therefore, they are expected to be used in multiple facets of nanomedicines and nano-formulations as they have their own innate therapeutic properties and they are relatively less toxic, safe, and biocompatible, with no known/reported adverse effects; therefore, their benefit to risk ratio is overwhelmingly positive overall.

## 11. Targeting Gene Regulation through the Use of Engineered EVs

The use of EVs for the targeted delivery of genetic material into cells is a promising area of investigation. The exosomal vesicles, loaded with specific microRNAs or CRISPR/Cas9, can regulate the expression of target genes, and studies have shown that this method can be effective. For example, grapefruit EVs loaded with siRNAs were able to effectively inhibit gene expression compared to other methods, and EVs isolated from ginger loaded with siRNA-CD98 effectively reduced the expression of the CD98 gene [82]. 

Studies on the delivery of the CRISPR/Cas9 system via EVs from tumor cells have shown promising results in inhibiting cancer cell proliferation and increasing cell sensitivity to chemotherapy drugs [96]. The benefits of EV-mediated delivery include low immunogenicity, ability to penetrate biological barriers, potential ability to reach and treat hard-to-reach areas such as solid tumors, and longer half-life in the blood compared to alternative methods [97,98,99]. EVs can be engineered to have specific proteins on their surfaces for more effective targeting, and they can also be loaded with the Cas9 ribonucleoprotein instead of a plasmid to avoid undesirable consequences [100]. 

All of these factors have the potential to make EV-based delivery methods a safer and more effective option for patients in the future.

## 12. Discussion on the Overall Commercial Viability of Plant-Derived Engineered EVs

PDEVs are signalosomes enclosed by a membrane that contain active biological material and have been linked to various functions. There is increasing evidence that they possess innate therapeutic properties, making them strong candidates for use in bio-nanomedicine. 

However, their interactions and effects with other hosts must be analyzed carefully to determine if they are specific or stochastic. More research on EV biology is necessary, as the biogenesis, functions, and uptake of EVs are not yet fully understood. Additionally, the identifying features of PDEVs are not clear, and more robust analytical methods are needed to identify their contents [15]. 

Isolating EVs is still an expensive process that requires complicated equipment and labor, and developing efficient isolation and scale-up methodologies is essential for translating their therapeutic potential. Clinical trials for a few PDEVs are currently underway, but regulatory issues (see Figure 3) surrounding their use as therapeutic agents remain unclear [12].

Despite these challenges, PDEVs offer several advantages in terms of biocompatibility, therapeutic ability, targeting capability, and cellular uptake. They are natural therapeutic agents without toxicity or side effects, which makes them highly desirable (see Table 1). With focused and multi-disciplinary expertise, PDEVs can be developed into reliable therapeutic agents for many common ailments.

## 13. Conclusions

PDEVs are important messengers between cells, carrying proteins, lipids, genetic material, and other bioactive components that can alter the expression of downstream cells. These EVs can not only transmit information within the same plant species but also exchange information with mammals across different kingdoms. In scientific research, PDEVs have shown a range of biological activities, such as anti-inflammation, anticancer, antioxidation, and antibacterial effects, among others. These EVs are biocompatible, biodegradable, and stable across a wide pH range, and they can be administered through various routes for disease treatment.

PDEVs have also shown promising results in the treatment of various diseases, such as colitis, cancer, alcohol-induced liver and kidney injury, and potentially even COVID-19. These EVs can serve as a drug delivery system, improving the bioavailability of small-molecule chemical drugs and nucleic acid drugs, while also reducing their side effects. The unique subcellular structures of PDEVs make them particularly effective carriers of small-molecule compounds and RNA.

Although interest in PDEVs is a relatively recent development, they have demonstrated a natural advantage for intercellular communication and a range of biological activities. Further studies are needed to standardize protocols for their isolation and characterization, develop storage technologies and efficient loading, and improve targeting of specific cell types. The regulation of plant nanovesicle properties through metabolic and genetic engineering of plant producers is particularly relevant to optimizing drug production. PDEVs have low immunogenicity, natural anti-inflammatory activity, and high biocompatibility, making them valuable contributors to the development of nanomedicine therapeutics.

This review concludes with the hope that it will enable researchers to be aware of the latest advances in PDEVs, specifically as its role as a nanocarrier of various drugs for the treatment of diseases, as growing evidence suggests that PDEVs will be the future in a variety of fields, especially in the pharmaceutical and biotechnological industries.

## 14. Recommendations—Formation of the PDEV Task Force

Based on the current state of the field and the need for further rigor and standardization in extracellular vesicle (EV) research, we propose the creation of a new task force focused on the standardization of plant-derived extracellular vesicles (PDEVs) by the International Society of Extracellular Vesicles (ISEV). PDEVs are an emerging area of research with tremendous potential in medicine, agriculture, and biotechnology. However, there is currently a lack of standardized methods for the collection, processing, and characterization of PDEVs, and currently no such task force exists to address concerns on a global scale.

The proposed PDEV Task Force would comprise experts in the field who can provide recommendations for improving reproducibility and standardization in PDEV research. The task force would focus on developing guidelines for the collection and processing of PDEVs, as well as methods for their characterization and functional analysis. The task force would also work to develop minimal reporting standards specific to PDEVs and identify gaps in current knowledge that require further investigation.

The recommendations of the PDEV Task Force would be disseminated as position papers, guidelines, minimal reporting standards, and online databases, and would be communicated to relevant societies and journals. As an ISEV member, you can propose joining the PDEV Task Force or other existing task forces by contacting the chair of the task force. ISEV aims to improve the reproducibility of EV research by funding research projects of task forces, working groups, and special interest groups. Researchers can apply for grants by completing an online application form.

We believe that the establishment of a PDEV Task Force will greatly benefit the field of EV research and facilitate the development of PDEVs as novel therapeutics and diagnostic tools. We hope our review will champion such an effort.

## Figures and Tables

**Figure 1 biomolecules-13-00839-f001:**
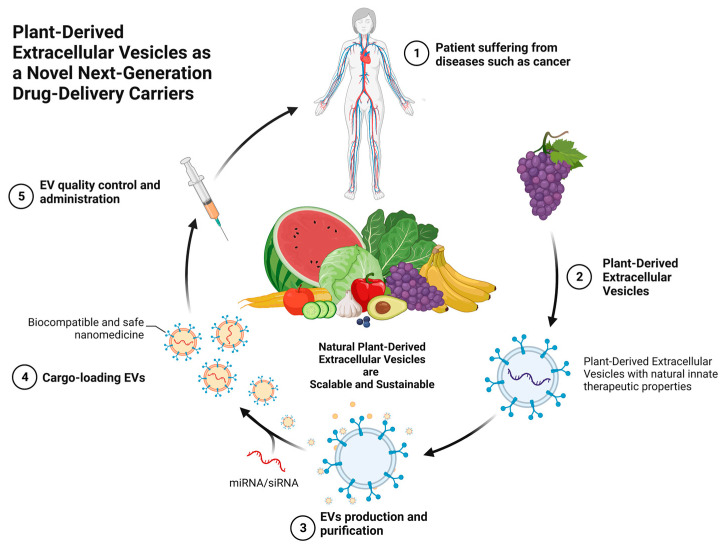
Graphical abstract: plant-derived extracellular vesicles as novel next-generation drug-delivery carriers. Stage 1: Patient suffering from diseases such as cancer that requires treatment; Stage 2: Plant-derived extracellular vesicles are isolated from plants; Stage 3: Production and purification to remove contaminants and pre-existing genetic material; Stage 4: Cargo-loading with drugs such as anticancer drugs; Stage 5: Comprehensive quality control and administration of cargo-loaded extracellular vesicles for treatment of the patient.

**Figure 2 biomolecules-13-00839-f002:**
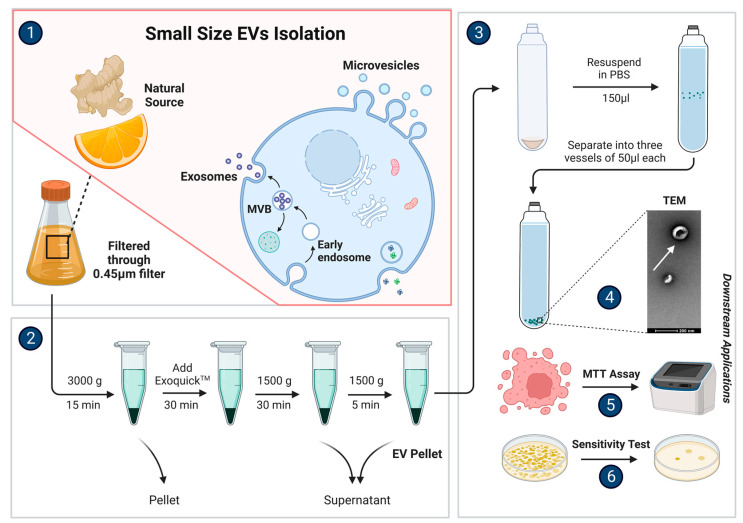
A method of extracellular vesicles isolation using commercial test kits that can be used for various downstream applications. Stage 1: Small-size EVs can be isolated from plant sources such as orange and ginger by filtering their juice through a 0.45 µm filter to remove larger debris and fibrous material; Stage 2: Thereafter, a series of low-speed centrifugation takes place to isolate the small-size EVs as per the manufacturer’s instructions; Stage 3: The sample is then resuspended in 1× PBS and can be separated based on the needs of the researchers and the requirements of the downstream applications; Stage 4: The sample can undergo physical characterization in order to ascertain whether the small-size EVs were indeed isolated and visualized using transmission electron microscopy (TEM); Stage 5: An example of a downstream application to test the natural chemotoxicity effect of the small-size EVs using an MTT assay; Stage 6: An example of a downstream application to test the antimicrobial effect of the small-size EVs by subjecting them to a sensitivity test.

**Figure 3 biomolecules-13-00839-f003:**
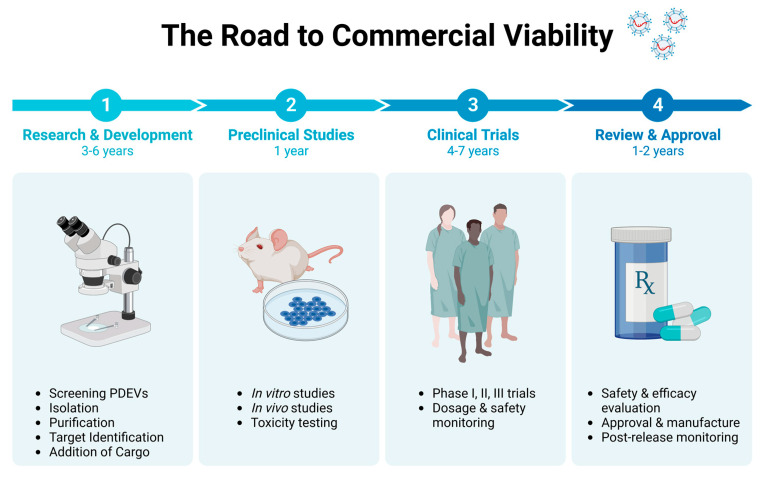
The road to commercial viability for PDEVs is a long and rigorous process. The data are based on the average estimates of commercial drug discovery and development processes [101,102].

**Table 1 biomolecules-13-00839-t001:** General advantages and disadvantages of PDEVs.

PDEVs	
Advantages	Disadvantages
Cross-kingdom communication	More extensive research is needed
Innate therapeutic properties	Currently expensive
Can be loaded with various cargoes	Currently labor-intensive
Targeting capability	Difficulties in isolation
Cellular uptake	Possibilities of impurities
Biocompatibility	Long road to commercial viability and approval
Ability to pass the blood-brain barrier	Regulatory compliance is still unknown
Protects fetus from medicines delivered to the mother because of their inability to pass through the placenta	
Less or no toxicity	
Safety	
No reported adverse effects	
Scalable and sustainable	

## Data Availability

Not applicable.

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
