# Peer review of "Plant-Derived Extracellular Vesicles and Their Exciting Potential as the Future of Next-Generation Drug Delivery"

_biomolecules, 2023, doi:10.3390/biom13050839_

Round 1
Reviewer 1 Report
This is a nice review on plant-derived EVs
First, it’s a pity that authors are not mentioning the Minimal Information for Studies of Extracellular Vesicles (MISEV) guidelines.
They are using the word “extracellular vesicles” which is appropriate; then “Plant-Derived Extracellular Vesicles (PDEVs)” (to be placed at line 16 instead of 18) but not using it along the text?
It should be the first keyword.
And I’m quite doubtful about “vegan” which is in fashion but is it the right place in a scientific paper, in my opinion?
Why the word:
- “exosome(s)” in figures?!
- “nanoparticles”, line 432
Authors should be cautious about:
- homogeneity in writing:
“Plant-derived EVs”, lines 15, 18…
“Plant derived EVs”, lines 39, 84, 333, 342, 575, 611…
“Interkingdom”, line 49
“Inter-kingdom”, line 97
“Grapefruit derived EVs”, lines 538-539, 542, 548
“cabbage derived EVs”, line 575
“Apple derived EVs”, line 578
- capital letters:
Why (?):
“Ginger derived miRNAs”, line 108
“Lipids”, line 115
Figure 2. “Method of Extracellular Vesicles Isolation using Commercial Test Kits”, line 221
“and Chinese”, line 260
”through Small RNA”, line 338
“treatment of Colitis”, line 401
“nanovesicles from Citrus”, line 416
“The Therapeutics Impact of Nanoparticles from Plants”, line 432
“The use of PDEVs for the Treatment of Alcoholic liver Disease”, line 490
“Capability of being able to Deliver Nucleic Acids”, line 516
“Capability of being able to Small Molecules and Drugs”, line 535
“Table 1. General Advantages”, line 658
“Cross-Kingdom Communication”
“Currently Labor intensive”
“Targeting Capability”
“Cellular Uptake”
“Scalable & Sustainable”
“The Road to Commercial Viability” in Figure 3
To be used:
“Nanoparticle Tracking Analysis (NTA)”, “Transmission Electron Microscopy (TEM)”, “Dynamic Light Scattering (DLS)”, lines 226-229
- italics and for all genders/species:
Arabidopsis thaliana, line 216
“in both in vivo and in vitro”, line 262
Lines 317, 319, 370-372, 420, 588
“via”, line 580
- bibliographic references:
15=55!
54=80!
Some other observations:
“a variety of plant and fruits”, line 106; a variety of plants and fruits?
“(plants or fruits)”, line 198; plants or fruits?
An extra space (“levels of functionally”), line 158?
“convenient method and easy method”, line 162; convenient and easy method?
I’m not sure blood plasma is the most commonly used source of EVs, line 211
“Stawberries-derived EVs”, line 285; Stawberry-derived EVs?
“gut microflora”, line 364; gut microbiota?
I don’t think that to call EVs “small bubbles”, line 503, is really scientific…
The cabbage-derived EVs are not naturally fluorescent, line 576; When PKH-labeled cabbage EVs were incubated ?
“with caco-2 cells”, line 578; Caco-2 cells
I recommend to publish this paper after major revision and in particular more attention paid to the manuscript and words, especially this vegan idea!
Reviewer 2 Report
The review by Alzahrani et al. describes the blooming use of plant-derived extracellular vesicles (PDEVs), either naturally loaded with plant bioactive molecules or used as carriers for exogenous drugs, in medicine. The problem is that the development of PDEV-based technologies is rapidly expanding, and so is the number of reviews on this topic, including several similar recent (March-April 2023) reviews appearing in a couple of MDPI journals, in Pharmaceutical Development and Technology, Cell Communication and Signaling, or Journal of Nanotechnology (doi: 10.1186/s12951-023-01858-7), and more. The job of a reviewer is then more difficult as I should compare the contents of those reviews with the presented manuscript. I tried to compare it at least with the review by Xu et al., from the Journal of Nanotechnology, as there is a large overlap in the content. The Alzahrani review focuses more on bowel inflammatory diseases, contains more detailed info on the characteristic features of the PDEVs, and I also like the introduction of the term “Vegan EVs”. I see it for the very first time in the literature, I hope it is indeed original. I generally like several conclusions and opinions provided by the authors. The review is broad, as reflected by the number of quoted references. The references include original papers, also several other reviews.
In my opinion, the manuscript contains many flaws, grammatical errors, and repetitions, and in its current form, it is not ready for publication. As this is a review, it can be (and should be) improved in a relatively short time (as other similar reviews can appear meanwhile). Although the whole manuscript should be carefully reviewed by the authors, the major corrections should include:
- English: it contains grammatical errors, do not use verb abbreviations, like "don’t, haven't", this is unacceptable in scientific papers.
- I don’t like this line-separated-paragraph layout, it is somehow distracting and pumping the volume of the ms. Many paragraphs could be merged.
- Please use latin names in Italics.
- Figure 1. Graphical Abstract – should have a legend. The arrow between stages 1 and 2 causes confusion, as it suggests this is a material isolated from a patient. I recommend removing the tube and vesicles and the arrow starting at the Patient suffering from a disease such as cancer, not making this schematic a closed circle. Better start an arrow going to stage 2 – Plant-derived EVs (instead of exosomes) from the plants in the middle of the schematic. Please change “exosomes” in the schematic for “EVs”
- Figure 2: It cannot use the heading: exosome isolation. Small size EVs isolation would be more true. The stages with numbers are missing descriptions?
- Size of EVs (line 36) “Their size from animal origin typically ranges from 30-150 nm [1]- please double check, this size corresponds with exosomes, animal EVs include microvesicles and apoptotic bodies, and their size range is usually described as 30 nm- 10 micrometer. I also think this reference 1 here is not a good one, as this is a review on plant-derived EVs, please quote a source for animal EVs. Also the size of PDEVs: the review by Nemati et al., from Cell Communication and Signaling provides several original references providing the size of many plant-derived EVs which are much larger than 500 nm, please check that.
- lines 39-40. I totally disagree with this hypothesis. The size of plant and animal EVs does not seem to differ that much, maybe then the authors should look for the average size of plant and animal EVs to support such a hypothesis.
- -please explain all the abbreviations when you use them for the first time.
- Nomenclature: there are many variations with the names of plant-derived extracellular vesicles, some researchers name them also EPDENs, etc., Maybe it is a good idea to be consistent at least in this review and avoid using the name “nanovesicles” instead of PDEVs?, e.g., in the 6.1.5 subtitle.
- I don’t understand the reasoning behind the structure of paragraph 6. It starts from a general introduction as 6: The Use of Plant-Originated EVs in Treating Human Illnesses, but then 6.1 with an almost identical title: 6.1. The Use of PDEVs for the Treatment of Various Diseases focuses only on small RNAs as cargo, suggesting that all the following subparagraphs 6.1.1. to 6.1.8 describe only the cases where the PDEVs were demonstrated to work via their sRNA cargo. Generally, 6.2 could be a separate paragraph as 7.
- Line 64: CRISPR-CAS9, should be Cas9.
- Line 64: Plant EVs can be scaled up economically and lines 70-71: The most significant limitation of EVs as a next-generation drug-delivery system is the need to generate a large-scale production of biocompatible EVs for commercial viability - seem to contradict, so can they be scaled up economically or is it problematic?
- Lines 97-100: Very unclear: Inter-kingdom communication and functionality of PDEVs have been attributed to miRNAs which have been shown to deliver functionally active miRNAs from dietary sources and are capable of surviving the harsh environment of the digestive tract [19]. miRNAs deliver miRNAs?
- line 257: As a result, further application of these drugs is limited [48]. – this reference is about inflammatory bowel disease and this paragraph describes the anticancer effect, re-think the reference. I also disagree with this final conclusion. It looks like all the anticancer drugs, including antibodies, are not good ones and there is a problem with their use.
- Table 1: Advantages and disadvantages: its layout is annoying, why are the columns so narrow, why is the text centered? Disadvantage: Not clear exactly how they interact with us -sounds unprofessional and it is unclear.
- Figure 3. The road to commercial viability for PDEVs is a long and rigorous process – is there a commercially available product that was used as an example for the time frame – what was the basis of the time frames for various stages of commercialization?
- Line 144 Describing ELD quoted in [31] as a method for PDEVs isolation as only “electrophoresis” can be misleading, it is electrophoresis combined with dialysis, I think the method should be described better.
- Line 302 and 304 repeat the same information: grapefruit-derived EVs activate the Wnt/TCF4 pathway
- Line 107: use “contain” instead of “have”
- Line 146; remove repeated “precipitation”
- line 472: this subtitle is grammatically incorrect. Please also explain what are Lgr5hi stem cells.
- line 492: “crucial hubs such as NRF2 and NLRP3”, explain what type of hubs, inflammasome?
- The subtitles in 6.2, starting as “Capability of being able to … “ are again grammatically incorrect, “Capability of delivering … would do”. In 6.2.1 I think it should be mentioned what are the methods of introducing exogenous nucleic acids to PDEVs.
- lines 539-540: contain repetition: siRNA, DNA, expression vectors, these are nucleic acids and as such should be mentioned in 6.2.2.
- line 551: MTX – please explain the abbreviation.
- line 578: caco-2 cells - should be Caco-2.
- References: Please include all the authors, instead of et al.
As regards Acknowledgments: with all respect for the authors’ beliefs, it is unprofessional. We acknowledge in writing people who contributed to the manuscript, e.g. by reading and comments.
English requires revision.
Round 2
Reviewer 1 Report
The manuscript has been significantly improved but it’s still not ready for publication.
- removing this “vegan” notion, “small bubbles”, and the “gratitude to Allah”,
- referring to MISEV, are really appreciated.
“Plant-Derived Extracellular Vesicles (PDEVs)” was chosen but already in the abstract, line 16, it’s written “Plant-derived EVs”; as at lines 33, 52, 792, 812…
With or without capital letters:
“Grapefruit-derived EVs”, line 56
“Mesenchymal stem cells”, line 86
“Blood plasma”, line 230
“Nanoparticle Tracking Analysis”, line 248
“The Adenosine Monophosphate activated Protein Kinase (AMPK), lines 550-551
“MTX-grapefruit-derived EVs”, line 678
“grapefruit-derived EVs”, line 680
In Figure 2, is a TEM image, not a NTA!
I recommend to publish this paper after minor revision with again more attention paid to the manuscript and words.
Reviewer 2 Report
The authors have thoroughly implemented my suggestions and comments and the review has been improved a lot. There are still minor remarks and corrections needed:
- Graphical abstract: I recommend placing the title in the legend. As regards the nucleic acid inside the PDEVs, I understand it symbolizes the intrinsic natural properties of the plant miRNA etc. But then the same nucleic acid symbol is added in stage 3 as miRNA/siRNA, which might be confusing. Maybe both nucleic acids should have different colors and after stage 3, both present in the EVs?
- Line 40-41: EVs generated from multivesicular bodies are exosomes, but then we have MVs, which are derived from the plasma membrane, so I recommend adding: generated through the multivesicular body pathway or from the plasma membrane to the extracellular space.
- Line 56: Grapefruit-grapefruit
- Line 58: They’re – they have
- Line 75: CAS9 – I would get back to CRISPR/Cas9 or CRISPR/Cas9 system as not only Cas9 enzyme is delivered via EVs.
- Lines 175-176 repeated offers in a single sentence, maybe change the second “offers” to “provides”? Lines 175-179 – generally this is a very long sentence, I recommend splitting it.
- Line 180-184: there are still some mistakes in this sentence: Test kits that are commercially available that and specialize in the isolation of EVs offers a convenient method and easy method to isolate EVs but suffers from low yields and low purity levels and they are generally used on a small scale [3], [38]. However, some manufacturers claim that their higher end kits are able to provide 420 times more EVs isolation compared to ultracentrifugation..
- Line 218: they’re – they are
- Line 230: Blood – blood
- As regards the new chapter on MISEV guidelines: I would like to know whether they include PDEVs, especially as regards category 2: collection and pre-processing – how the plant sources of EVs can be standardized. If MISEV2018 does not deal with plant origin, maybe the authors could briefly suggest which features of plant resources should in their opinion be provided to standardize the PDEV material.
- Line 331: remove the first “and”
- Line 414: the PA abbreviation is not clear here, maybe use the full name?
- Lines 667-680 – there is no need to used capitals with grapefruit-derived EVs inside a sentence.
English level has been imroved, there are still small errors that can be corrected.
